# Controlled Germination of Faba Beans: Drying, Thermodynamic Properties and Physical-Chemical Composition

Lumara Tatiely Santos Amadeu [1], Alexandre José de Melo Queiroz [1,*], Rossana Maria Feitosa de Figueirêdo [1], João Paulo de Lima Ferreira [1], Wilton Pereira da Silva [1], Josivanda Palmeira Gomes [1], Yaroslávia Ferreira Paiva [2], Caciana Cavalcanti Costa [3], Henrique Valentim Moura [1], Dyego da Costa Santos [4], Ana Raquel Carmo de Lima [5] and Hanndson Araujo Silva [2]

1   Department of Agricultural Engineering, Federal University of Campina Grande, Campina Grande 58429-900, Brazil; lumaratatielyea@gmail.com (L.T.S.A.); rossanamff@gmail.com (R.M.F.d.F.); joaop_l@hotmail.com (J.P.d.L.F.); wiltonps@uol.com.br (W.P.d.S.); josivanda@gmail.com (J.P.G.); valentim_henrique@hotmail.com (H.V.M.)
2   Science and Technology Center, Federal University of Campina Grande, Campina Grande 58429-900, Brazil; yaroslaviapaiva@gmail.com (Y.F.P.); hanndson@gmail.com (H.A.S.)
3   Center for Sustainable Development of the Semiarid, Federal University of Campina Grande, Sumé 58540-000, Brazil; caciana.cavalcanti@professor.ufcg.edu.br
4   Department of Technology in Agroindustry, Federal Institute of Education, Science and Technology of Rio Grande do Norte, Pau dos Ferros 59900-000, Brazil; dyego.csantos@gmail.com
5   Department of Technology in Agroindustry, Federal Institute of Education, Science and Technology of de Alagoas, Batalha 57420-000, Brazil; ana.carmo@ifal.edu.br
*   Correspondence: alexandre.melo@professor.ufcg.edu.br

**Abstract:** The objective of this work was to determine the drying kinetics and the thermodynamic properties of the drying process of germinated seeds from faba beans of the Olho-de-Vó Preta (OVP), Raio-de-Sol (RS) and Branca (B) varieties. Additionally, the physicochemical properties of the germinated seeds and subsequent dried flours were determined. A thin layer of seeds were dried using a convective dryer at temperatures of 50, 60, 70 and 80 °C. Mathematical models were applied to the drying experimental data. The samples were further characterized for water content, water activity, ash, pH, alcohol-soluble acidity, total and reducing sugars, proteins, and starch. Page and Midilli models revealed the best predictions of the drying kinetics for all evaluated conditions. The effective diffusion coefficient increased with increasing temperature and presented magnitude in the order of $10^{-9}$ m$^2$/s. The activation energy presented results in the range of 19 and 27 kJ/mol, falling within the range reported for agricultural products. The entropy and enthalpy values were higher in the OVP, followed by RS, higher than in the B variety. The increase in drying temperature resulted in a reduction of enthalpy and entropy and an increase in Gibbs free energy, indicating that the drying process is endothermic and requires external energy. Samples have acidic pH and acidity decreased with drying; the RS and B varieties had higher sugar contents; the B variety had the highest protein contents, and these were obtained from the *in natura* germinated samples; in the B variety the highest starch content was obtained. All flours showed good characteristics, presenting themselves as an alternative for diversifying the supply of beans.

**Keywords:** *Phaseolus lunatus* L.; malting; drying kinetics; mathematical modeling; flour; nutrition





## 1. Introduction

The faba bean (*Phaseolus lunatus* L.) is also known as the lima bean [1], which is a legume grown mainly in the Northeast region of Brazil. Its production corresponded to 11,381 tons in 2019, in 36,252 hectares, with the states of Ceará (4.614 t), Paraiba (2.910 t) and Rio Grande do Norte (1.274 t) being the three largest producers in Brazil [2]. Despite being a culture considered for subsistence, it presents significant commercial exploitation due to the high commercial value of the grains and is considered the second most consumed bean

species in Brazil [3,4]. The faba bean seeds are high in protein (16 to 21%) and carbohydrates (55 to 61%), low in fat (1 to 2.3%), having fiber levels from 3.2 to 6.8%, elevated levels of minerals such as K, Zn, Ca and Fe and low levels of Na and P [5]. The beans contain protease inhibitors, which help fight the development of cancer cells [6].

With the growing trend in the search for healthy and nutritious products, techniques that increase the nutritional value of foods at a low cost are of permanent interest. The improvement of the nutritional value of seeds can be done through the controlled use of the germination process, which conveniently alters the quality of the original raw material, increasing commercial potential and application possibilities. After germination, the seeds have high water content, a factor that compromises quality and confers high perishability to the product, requiring prior treatment that prevents rapid deterioration, enabling further processing or commercialization.

Among the technologies that enable the extension of the useful life of grains and seeds, convective drying is the most widely used, offering attractive costs and operational simplicity. Convective drying is one of the most used technologies for the preservation of agricultural products, combining benefits of increased shelf life, weight, and volume reduction, consequently reducing costs with packaging, transport and storage. Furthermore, drying can preserve the nutritional quality of faba bean, while providing easy availability to an increased added value product [7]. However, the inadequate use of convective drying can harm the physical, sensory, and nutritional quality of the product. Thus, the drying process should be planned accordingly to avoid damage, serving the control and administration of various stages of agribusiness [8,9].

The behavior of a product subjected to a dehydration process is evaluated using drying kinetics, which involves different conditions such as temperature and air velocity [10]. The drying process can be represented by mathematical modeling, helping to improve and design equipment, in addition to providing information for the optimization of the process for each raw material [11,12]. Beyond the drying kinetics, the determination of thermodynamic properties plays a significant role in the process, as it helps to calculate the energy needed to remove the moisture from the sample. These properties help with evaluating the physical phenomena that occurs on the surface of the products and contribute to the scale-up of the process [13]. Studies on drying kinetics deserve interest due to the different biological structures involved in the heat and mass transfer of each product [14].

Even though several studies related to bean culture can be found, different varieties still lack work. Additionally works related to the convective drying of germinated seeds are scarce, including faba bean varieties. Therefore, the objective of our work was to determine the drying kinetics of germinated faba bean seeds, from the varieties Olho-de-Vó, Raio-de-Sol and Branca at temperatures of 50, 60, 70 and 80 °C, to determine the thermodynamic properties of the process and to characterize physico-chemical properties of the germinated seeds and subsequent dried flours.

## 2. Materials and Methods

### 2.1. Material

Three varieties (Orelha-de-Vó Preta, Raio-de-Sol and Branca) of faba bean seeds (*Phaseolus lunatus* L.) produced in the region of the city of Campina Grande were used (Geographical coordinates: 7°13′50″ S and 35°52′52″ O), Paraiba, Brazil. The seeds were received and hand-picked, choosing the intact ones and with uniform size.

### 2.2. Processing and Germination of Seeds

About 100 g of seeds of each faba bean variety were sanitized by immersion in a 7% (*w*/*v*) sodium hypochlorite solution at a 1:10 ratio (seeds/solution) at room temperature (28 ± 1 °C) for 5 min. Soon after sanitization, the seeds were washed with distilled water, drained, and placed in trays at room temperature to eliminate surface water. The seeds were germinated following the recommendations of the Rules for Seed Analysis [15], using germitest paper. Fifty-six seeds were distributed per leaf, kept in BOD-type chambers at

25 °C, for 72 h (Olho-de-vó preta) and 96 h (Raio-de-Sol and Branca). Germination times were chosen based on previous studies [16,17] and preliminary germination tests (data not shown). The seeds were irrigated every 24 h, applying about 25 mL of distilled water.

### 2.3. Drying Procedure

The germinated faba bean seeds were crushed in a domestic blender and spread on stainless steel screened trays, in a thin layer, with an approximate height of 0.64 cm. The samples were subjected to drying in a convective dryer at temperatures of 50, 60, 70 and 80 °C and drying air speed of 1.0 m/s. The dryings were performed in triplicate, with the samples being weighed on an analytical balance with a precision of 0.0001 g, at regular times of 5, 10, 20, 30 and 60 min, until a constant mass was obtained. The water content at the end of the drying kinetics was determined gravimetrically by drying in an oven at 105 °C for 24 h [18].

### 2.4. Data Modeling

The drying data were used after converting the moisture loss data into the dimensionless moisture content ratio (MR) parameter, according to Equation (1).

$$MR = \frac{M_t - M_e}{M_i - M_e} \tag{1}$$

where, MR is the moisture content ratio (dimensionless); M is the moisture content (% dry basis); $M_e$ is the equilibrium moisture content (% dry basis); $M_0$ is the initial moisture content (% dry basis). Then, the mathematical models presented in Table 1 were fitted to the drying kinetics data, represented by the ratio of moisture content as a function of drying time, using the computer program Statistica version 7.0 (StatSoft® Inc., Tulsa, OK, USA) using regression nonlinear and the Quasi-Newton method.

**Table 1.** Mathematical models used to estimate the drying kinetics curves of the faba beans.

| Model Name | Equation | | References |
|---|---|---|---|
| Newton | $MR = \exp(-kt)$ | (2) | [19] |
| Page | $MR = \exp(-kt^n)$ | (3) | [20] |
| Henderson and Pabis | $MR = a \exp(-kt)$ | (4) | [21] |
| Modified Henderson and Pabis | $MR = a \exp(-kt) + b\exp(-k_0t) + c \exp(-k_1t)$ | (5) | [22] |
| Thompson | $MR = \exp(-a - (a^2 + 4bt)^{0,5})/2b$ | (6) | [23] |
| Logarithmic | $MR = a \exp(-kt) + c$ | (7) | [24] |
| Two terms | $MR = (-k_0t) + b \exp(-k_1t)$ | (8) | [25] |
| Midilli | $MR = a \exp(-kt^n) + bt$ | (9) | [26] |
| Approximation of Diffusion | $MR = a \exp(-kt) + (1-a)\exp(-kbt)$ | (10) | [27] |
| Two-term exponential | $MR = a \exp(-kt) + (1-a)\exp(-kat)$ | (11) | [27] |
| Verma | $MR = a \exp(-kt) + (1-a)\exp(-k_1at)$ | (12) | [28] |

MR—moisture content ratio (dimensionless); a, b, c, k, k0, k1, n—model parameters; t—drying time (min).

The mathematical models were evaluated for the quality of the adjustments, taking as parameters the magnitude of the coefficient of determination ($R^2$), the mean square deviation (MSD) (Equation (13)) and the chi-square ($\chi^2$) (Equation (14)).

$$MSD = \left[ \frac{1}{N} \sum_{i=1}^{N} \left( MR_{pred,i} - MR_{exp,i} \right)^2 \right]^{\frac{1}{2}} \tag{13}$$

$$\chi^2 = \frac{1}{N-n} \sum_{i=1}^{N} \left( MR_{pred,i} - MR_{exp,i} \right)^2 \tag{14}$$

where, $MR_{pred}$ is the moisture content ratio predicted by the model; $MR_{exp}$ is the experimental moisture content ratio; N is the number of observations; n is the number of model constants.

### 2.5. Determination of the Effective Diffusion Coefficient and Activation Energy

The effective diffusion coefficients were determined by fitting the mathematical model of liquid diffusion, with approximation of four terms (Equation (15)), to the experimental data of the drying kinetics, considering the uniform initial moisture distribution, constant diffusivity and external resistance and negligible volume contraction. This model is the analytical solution of Fick's second law considering the geometric form of the material as approximated to a flat plate [29].

$$MR = \frac{M_t - M_e}{M_i - M_e} = \frac{8}{\pi^2} \sum_{n=0}^{\infty} \frac{1}{(2n+1)^2} \exp\left[-(2n+1)^2\pi^2 D_{ef}\frac{t}{4L^2}\right] \tag{15}$$

where, $D_{ef}$ is the effective diffusion coefficient ($m^2/s$); n is the number of terms in the equation; L is the characteristic dimension (half sample thickness) (m); t is the time(s).

The influence of temperature on the effective diffusion coefficients was evaluated using an Arrhenius-type equation (Equation (16)).

$$D_{ef} = D_0\left(-\frac{E_e}{RT}\right) \tag{16}$$

where, $D_0$ is the pre-exponential factor ($m^2/s$); $E_a$ is the activation energy (kJ/mol); R is the universal gas constant (0.008314 kJ/mol K); T is the absolute temperature (K).

### 2.6. Determination of the Thermodynamic Properties

The thermodynamic properties of enthalpy (Equation (17)), entropy (Equation (18)) and Gibbs free energy (Equation (19)) in the drying process of germinated faba bean seeds were quantified using the equations described by Silva et al. [30].

$$\Delta H = E_a - RT \tag{17}$$

$$\Delta S = R\left[\ln D_0 - \ln\left(\frac{k_b}{h_p}\right) - \ln(T)\right] \tag{18}$$

$$\Delta G = \Delta H - T\Delta S \tag{19}$$

where, $\Delta H$ is the specific enthalpy (J/mol); $\Delta S$ is the specific entropy (J/mol K); $\Delta G$ is the Gibbs free energy (J/mol); Kb is the Boltzmann constant ($1.38 \times 10^{-23}$ J/K); hp is the Planck's constant ($6.626 \times 10^{-34}$ J/s); T is the absolute temperature (K).

### 2.7. Physicochemical Characterization

The germinated seeds and the flours from the dried germinated seeds were characterized, in triplicate, for each physicochemical property. According to the analytical procedures of Adolfo Lutz Institute [18], the moisture content was determined by the gravimetric method in an oven at 105 °C/24 h; alcohol-soluble acidity, by titration with 0.1 M NaOH; pH determined in digital potentiometer; ash content, by incineration in a muffle at 550 °C. Water activity was determined at 25 °C by direct reading using an Aqualab meter (3TE model, Decagon Devices, São José dos Campos, Brazil). The reducing sugars were determined by the method of dinitrosalicylic acid [31]; total sugars by the anthrone method [32]; starch by the methodology of the Adolf Lutz Institute [18], which is based on the acid hydrolysis of starch and on the titration of the Fehling solution; and the protein content, quantified according to the Kjeldahl method, where the total nitrogen content is determined, with the total protein being determined by multiplying the total nitrogen content by a factor of 6.25.

### 2.8. Statistical Analysis

The results of the physicochemical analyses were expressed as the mean ± standard. The data analysis was performed in a completely randomized design and the differences between treatment means were determined using one-way analysis of variance (ANOVA) and applying the Tukey test at 5% probability, using Assistat software version 7.7 Beta (Federal University of Campina Grande, Campina Grande, Paraíba, Brazil) [33].

## 3. Results

### 3.1. Mathematical Modeling of Drying Kinetics

Table 2 shows the coefficients of determination ($R^2$), the mean square deviations (MSD) and the chi-squares ($\chi^2$) obtained for each mathematical model adjusted to the drying kinetics of the germinated seeds of faba beans, varieties Orelha-de-Vó Preta (OVP), Raio-de-Sol (RS) and Branca (B), at temperatures of 50, 60, 70 and 80 °C. It is observed that for the three varieties under study, all mathematical models presented $R^2 > 0.9900$, MSD $\leq 0.040$ and $\chi^2 \leq 13.2843 \times 10^{-4}$. For a model to satisfactorily represent a drying process, it is essential that the coefficient of determination ($R^2$) is greater than 0.99, and that the mean square deviations (MSD) and chi-squares ($\chi^2$) have the lowest possible values [34]. It is observed that all models, as they present $R^2$ above 0.990 and values close to zero for MSD and $\chi^2$, can be used to represent the behavior under drying from 50 to 80 °C of the three varieties of germinated seeds of faba beans. However, of the 11 models assessed, the Page and Midilli models stood out for presenting the highest $R^2$ ($\geq 0.9997$) and the lowest MSD ($\leq 0.0064$) and $\chi^2$ ($\leq 0.4647 \times 10^{-4}$), demonstrating excellent fits to the experimental data.

**Table 2.** Coefficients of determination ($R^2$), mean square deviations (MSD) and chi-squares ($\chi^2$) of the mathematical models adjusted to the drying kinetics data of the germinated seeds of faba beans (varieties OVP, RS, B), at temperatures of 50, 60, 70 and 80 °C.

| Models | T (°C) | OVP | | | RS | | | B | | |
|---|---|---|---|---|---|---|---|---|---|---|
| | | $R^2$ | MSD | $\chi^2$ | $R^2$ | MSD | $\chi^2$ | $R^2$ | MSD | $\chi^2$ |
| Newton | 50 | 0.9991 | 0.0119 | 1.4696 | 0.9984 | 0.0165 | 2.8440 | 0.9957 | 0.0262 | 7.1549 |
| | 60 | 0.9975 | 0.0193 | 3.8834 | 0.9982 | 0.0162 | 2.7445 | 0.9953 | 0.0270 | 7.5771 |
| | 70 | 0.9980 | 0.0176 | 3.2418 | 0.9975 | 0.0190 | 3.7827 | 0.9961 | 0.0241 | 6.0580 |
| | 80 | 0.9970 | 0.0204 | 4.3462 | 0.9968 | 0.0212 | 4.6884 | 0.9940 | 0.0302 | 9.5150 |
| Page | 50 | 0.9999 | 0.0040 | 0.1757 | 0.9999 | 0.0037 | 0.1518 | 0.9999 | 0.0038 | 0.1585 |
| | 60 | 0.9999 | 0.0048 | 0.2529 | 0.9999 | 0.0041 | 0.1823 | 0.9998 | 0.0054 | 0.3176 |
| | 70 | 0.9998 | 0.0058 | 0.3726 | 0.9998 | 0.0055 | 0.3307 | 0.9997 | 0.0064 | 0.4415 |
| | 80 | 0.9997 | 0.0060 | 0.3912 | 0.9997 | 0.0064 | 0.4505 | 0.9998 | 0.0062 | 0.4183 |
| Henderson and Pabis | 50 | 0.9994 | 0.0096 | 0.9975 | 0.9989 | 0.0124 | 1.6673 | 0.9970 | 0.0203 | 4.4385 |
| | 60 | 0.9981 | 0.0156 | 2.6597 | 0.9987 | 0.0130 | 1.8245 | 0.9965 | 0.0219 | 5.2130 |
| | 70 | 0.9984 | 0.0157 | 2.7129 | 0.9980 | 0.0171 | 3.1914 | 0.9969 | 0.0216 | 5.0900 |
| | 80 | 0.9974 | 0.0188 | 3.8938 | 0.9974 | 0.0192 | 4.0333 | 0.9952 | 0.0271 | 7.9877 |
| Modified Henderson and Pabis | 50 | 0.9994 | 0.0096 | 1.2073 | 0.9989 | 0.0124 | 2.0007 | 0.9970 | 0.0203 | 5.2840 |
| | 60 | 0.9981 | 0.0156 | 3.2856 | 0.9987 | 0.0130 | 2.2086 | 0.9965 | 0.0219 | 6.2556 |
| | 70 | 0.9984 | 0.0157 | 3.3512 | 0.9980 | 0.0171 | 3.9006 | 0.9969 | 0.0216 | 6.2211 |
| | 80 | 0.9974 | 0.0188 | 4.8673 | 0.9974 | 0.0192 | 4.9823 | 0.9952 | 0.0271 | 9.7628 |
| Thompson | 50 | 0.9974 | 0.0207 | 4.6441 | 0.9981 | 0.0174 | 3.2902 | 0.9953 | 0.0280 | 8.4414 |
| | 60 | 0.9970 | 0.0221 | 5.3326 | 0.9974 | 0.0206 | 4.6071 | 0.9928 | 0.0350 | 13.2843 |
| | 70 | 0.9979 | 0.0180 | 3.5467 | 0.9973 | 0.0196 | 4.1802 | 0.9959 | 0.0247 | 6.6588 |
| | 80 | 0.9927 | 0.0315 | 10.9041 | 0.9957 | 0.0247 | 6.6753 | 0.9924 | 0.0339 | 12.5472 |

**Table 2.** *Cont.*

| Models | T (°C) | OVP | | | RS | | | B | | |
|---|---|---|---|---|---|---|---|---|---|---|
| | | R² | MSD | χ² | R² | MSD | χ² | R² | MSD | χ² |
| Logarithmic | 50 | 0.9994 | 0.0096 | 1.0440 | 0.9989 | 0.0124 | 1.7365 | 0.9971 | 0.0200 | 4.5178 |
| | 60 | 0.9982 | 0.0155 | 2.7747 | 0.9987 | 0.0129 | 1.9020 | 0.9967 | 0.0216 | 5.2967 |
| | 70 | 0.9984 | 0.0157 | 2.8464 | 0.9980 | 0.0171 | 3.3336 | 0.9969 | 0.0214 | 5.2362 |
| | 80 | 0.9974 | 0.0188 | 4.0958 | 0.9974 | 0.0192 | 4.2343 | 0.9953 | 0.0268 | 8.2042 |
| Two terms | 50 | 0.9994 | 0.0096 | 1.0925 | 0.9989 | 0.0124 | 1.8189 | 0.9970 | 0.0203 | 4.8277 |
| | 60 | 0.9981 | 0.0156 | 2.9397 | 0.9987 | 0.0130 | 1.9983 | 0.9965 | 0.0219 | 5.6874 |
| | 70 | 0.9984 | 0.0157 | 2.9984 | 0.9980 | 0.0171 | 3.5105 | 0.9969 | 0.0216 | 5.5990 |
| | 80 | 0.9974 | 0.0188 | 4.3265 | 0.9974 | 0.0192 | 4.4578 | 0.9952 | 0.0271 | 8.7865 |
| Midilli | 50 | 0.9999 | 0.0037 | 0.1628 | 0.9999 | 0.0033 | 0.1315 | 0.9999 | 0.0035 | 0.1467 |
| | 60 | 0.9999 | 0.0046 | 0.2511 | 0.9999 | 0.0039 | 0.1771 | 0.9998 | 0.0050 | 0.2922 |
| | 70 | 0.9998 | 0.0055 | 0.3663 | 0.9998 | 0.0053 | 0.3409 | 0.9997 | 0.0062 | 0.4647 |
| | 80 | 0.9998 | 0.0057 | 0.3928 | 0.9997 | 0.0061 | 0.4523 | 0.9998 | 0.0060 | 0.4252 |
| Approximation of Diffusion | 50 | 0.9999 | 0.0037 | 0.1547 | 0.9999 | 0.0033 | 0.1233 | 0.9999 | 0.0038 | 0.1615 |
| | 60 | 0.9999 | 0.0044 | 0.2193 | 0.9999 | 0.0038 | 0.1599 | 0.9998 | 0.0059 | 0.3987 |
| | 70 | 0.9998 | 0.0055 | 0.3429 | 0.9983 | 0.0157 | 2.8032 | 0.9976 | 0.0191 | 4.1524 |
| | 80 | 0.9970 | 0.0204 | 4.8037 | 0.9977 | 0.0180 | 3.7214 | 0.9960 | 0.0247 | 6.9530 |
| Two-term exponential | 50 | 0.9991 | 0.0122 | 1.6292 | 0.9983 | 0.0169 | 3.0821 | 0.9956 | 0.0265 | 7.5949 |
| | 60 | 0.9974 | 0.0195 | 4.1739 | 0.9981 | 0.0165 | 2.9612 | 0.9952 | 0.0273 | 8.0462 |
| | 70 | 0.9979 | 0.0180 | 3.5331 | 0.9975 | 0.0190 | 3.9547 | 0.9960 | 0.0244 | 6.4809 |
| | 80 | 0.9998 | 0.0059 | 0.3800 | 0.9997 | 0.0061 | 0.4106 | 0.9939 | 0.0305 | 10.1270 |
| Verna | 50 | 0.9999 | 0.0037 | 0.1547 | 0.9999 | 0.0033 | 0.1233 | 0.9957 | 0.0262 | 7.7511 |
| | 60 | 0.9975 | 0.0193 | 4.2718 | 0.9982 | 0.0162 | 2.9940 | 0.9953 | 0.0270 | 8.2360 |
| | 70 | 0.9980 | 0.0176 | 3.5660 | 0.9975 | 0.0190 | 4.1430 | 0.9961 | 0.0241 | 6.6349 |
| | 80 | 0.9979 | 0.0170 | 3.3346 | 0.9968 | 0.0212 | 5.1572 | 0.9940 | 0.0302 | 10.4212 |

OVP—Orelha-de-Vó Preta; RS—Raio-de-Sol; B—Branca.

Among the models that presented the best fits, although Midilli had the smallest mean squared deviations and reduced chi-squares, the Page model, being simpler and using only two parameters, simplifying its application in mathematical simulations, is the recommended model to represent drying in a thin layer of the germinated faba bean seeds. The Midilli and Page models were also reported as satisfactory for estimating the drying kinetic curves by Lisboa et al. [35] for mulatto beans (*Phaseolus vulgaris* L.), at temperatures of 40, 50, 60 and 70 °C, with drying air speed of 1.0 m/s; and by Rahmanian-Koushkaki et al. [36] for corn seeds, at temperatures of 40, 50 and 60 °C, in which among the models tested, the Page was the most suitable. Figure 1 shows the experimental points and curves estimated by the Page model for the ratio of water content as a function of drying time for germinated seeds of faba beans of the OVP, RS and B varieties, at temperatures of 50, 60, 70 and 80 °C.

The best use of the energy spent on drying is observed to be the period between 50 to 100 min, with moisture content ratios close to zero from 100 min onwards, even at a temperature of 50 °C. According to Zielinska and Michalska et al. [37] this reflects the high-moisture content available under weak molecular binding and as the process progressed, drying rates decreased, possibly due to increased internal resistance to heat and mass transfer. This behavior was also reported by Chielle et al. [38] on papaya seeds (*Carica papaya* L.), Hasan et al. [39] on rice seeds. Increasing the temperature increases the difference between the vapor pressure of the drying air and the samples, therefore, higher temperatures result in greater and faster water removal, as observed in watermelon seeds [40], canola [41] and common beans [42].

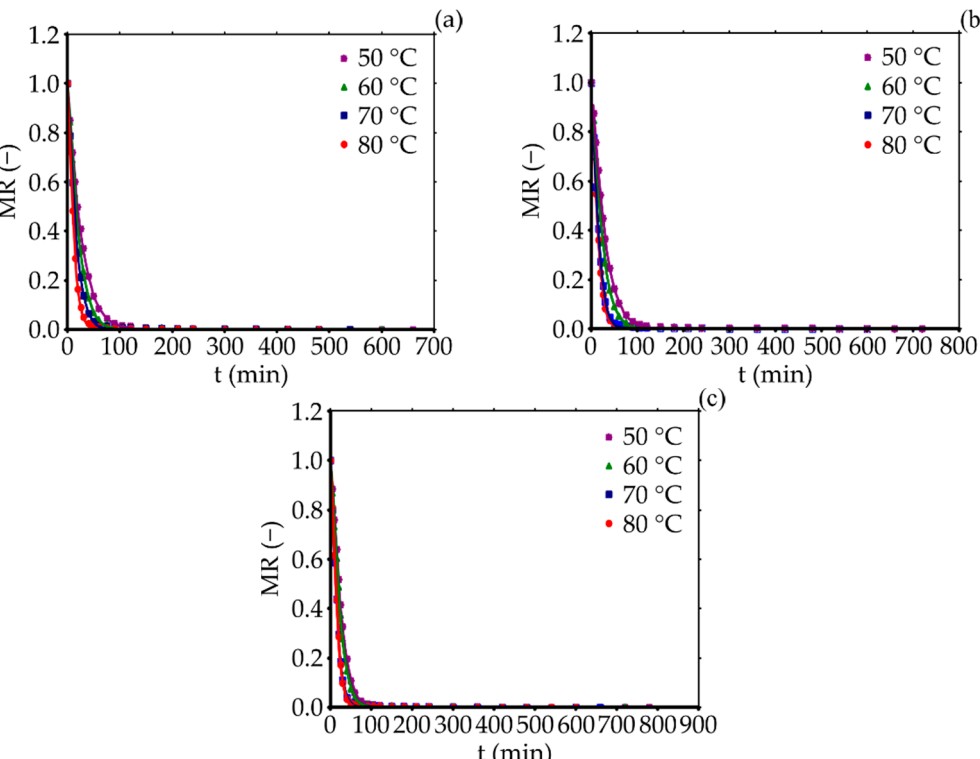

**Figure 1.** Experimental and estimated values by the Page model for the ratio of water content as a function of drying time of germinated faba bean seeds at temperatures of 50, 60, 70 and 80 °C: (**a**) OVP; (**b**) RS; and (**c**) B.

The average time required to complete the drying process of the 'Orelha-de-Vó' variety (OVP) ranged from 480 to 660 min for temperatures between 50 and 80 °C, for the 'Raio-de-Sol' variety (RS) the process lasted between 540 and 720 min and, finally, in the 'Branca' variety (B), between 600 and 720 min. Similar drying times were verified by Ferreira et al. [43] when analyzing the drying kinetics of germinated pumpkin seeds, which reported drying times between 470 min (70 °C) and 720 min (50 °C). Lisboa et al. [35], in the mathematical description of the drying curves of mulatto beans (*Phaseolus vulgaris* L.), observed an average time of 1300, 1000, 880 and 640 min to complete the drying process at temperatures of 40, 50, 60 and 70 ° C, respectively.

In Figure 2, we have the representation of the experimental moisture content ratio values and the values predicted by the Page model. The good prediction, represented by the curve, is verified by its superposition with the experimental points determined in the drying kinetics, corroborating the satisfactory results of $R^2$, MSD and $\chi^2$.

Table 3 shows the parameters of the Page model adjusted to the drying kinetics data of germinated seeds of faba beans of different varieties at temperatures of 50, 60, 70 and 80 °C OVP and intermediate values in the RS. For the drying parameter "k", the lowest values, while comparing the same temperatures, were found for the Branca variety, and the highest in the OVP, but close to RS. According to Lisboa et al. [35], this constant represents the effect of external drying conditions, indicating that the drying rate increases with an increase in air temperature. The authors reported similar behavior to those observed here when drying mulatto beans (*Phaseolus vulgaris* L.) at temperatures of 40, 50, 60 and 70 °C, on what "k" and "n" showed as an increase as the temperature was increased.

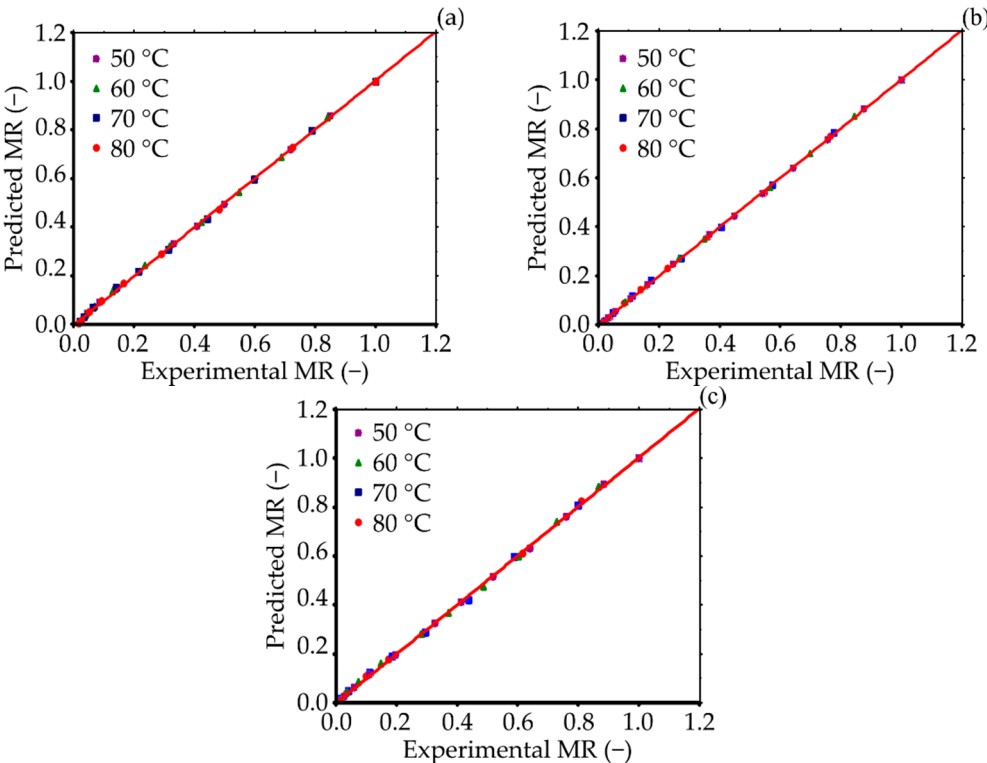

**Figure 2.** Relationship between the values predicted by the Page model and the experimental water content ratio values in the drying of germinated faba bean seeds at temperatures of 50, 60, 70 and 80 °C: (**a**) OVP; (**b**) RS; and (**c**) B.

**Table 3.** Parameters of the Page model adjusted to experimental data of drying kinetics of germinated seeds of faba bean varieties, at temperatures of 50, 60, 70 and 80 °C.

| Variety | T (°C) | k | n |
|---|---|---|---|
| OVP | 50 | 0.0257 | 1.1067 |
| | 60 | 0.0233 | 1.2085 |
| | 70 | 0.0339 | 1.1829 |
| | 80 | 0.0426 | 1.2444 |
| RS | 50 | 0.0194 | 1.1591 |
| | 60 | 0.0242 | 1.1731 |
| | 70 | 0.0342 | 1.2149 |
| | 80 | 0.0351 | 1.2453 |
| B | 50 | 0.0138 | 1.2923 |
| | 60 | 0.0151 | 1.3030 |
| | 70 | 0.0273 | 1.2749 |
| | 80 | 0.0211 | 1.3669 |

OVP—Orelha-de-Vó Preta; RS—Raio-de-Sol; B—Branca.

### 3.2. Diffusion Coefficient and Activation Energy

Table 4 shows the average effective diffusion coefficients ($D_{ef}$) obtained from drying germinated faba beans at temperatures of 50, 60, 70 and 80 °C. It is observed that the values of the diffusion coefficients ranged from 1.7890 to 4.6411 $\times 10^{-9}$ m$^2$/s, for the varieties RS at 50 °C and OVP at 80 °C, respectively, lying within the range mentioned by Madamba et al. [44] for food products from $10^{-11}$ to $10^{-9}$ m$^2$/s. The increase in temperature caused the increase in the diffusion coefficient. This behavior can be explained by the greater agitation of the water molecules, which reduces their attraction forces and their resistance to flow, thus facilitating the diffusion of water to the surface of the sample [45,46]. During the drying of Gandu beans (*Cajanus cajan* (L.) Mills.), Silva et al. [42]

reported effective diffusivity with values between $2.1 \times 10^{-10}$ and $6.8 \times 10^{-10}$ m$^2$/s, for a range between 40 and 70 °C, demonstrating an increase with the increase in drying air temperature, as observed for the germinated seeds of faba beans.

**Table 4.** Effective diffusion coefficients obtained from drying germinated seeds of faba beans at temperatures of 50, 60, 70 and 80 °C.

| Variety | T (°C) | $D_{ef} \times 10^{-9}$ (m$^2$/s) | $R^2$ |
|---|---|---|---|
| OVP | 50 | 1.9763 | 0.9887 |
| | 60 | 2.4541 | 0.9849 |
| | 70 | 3.2291 | 0.9874 |
| | 80 | 4.6411 | 0.9851 |
| RS | 50 | 1.7890 | 0.9860 |
| | 60 | 2.2939 | 0.9865 |
| | 70 | 3.5491 | 0.9868 |
| | 80 | 3.9182 | 0.9859 |
| B | 50 | 1.9761 | 0.9809 |
| | 60 | 2.2026 | 0.9810 |
| | 70 | 3.4098 | 0.9841 |
| | 80 | 3.4303 | 0.9805 |

OVP—Orelha-de-Vó Preta; RS—Raio-de-Sol; B—Branca; $R^2$—coefficient of determination.

The linearized moisture diffusion coefficients were plotted as a function of the inverse of the absolute drying temperature (Figure 3), and its dependence on the drying air temperature was satisfactorily represented by an Arrhenius type equation, which presented values of $R^2 \geq 0.8783$.

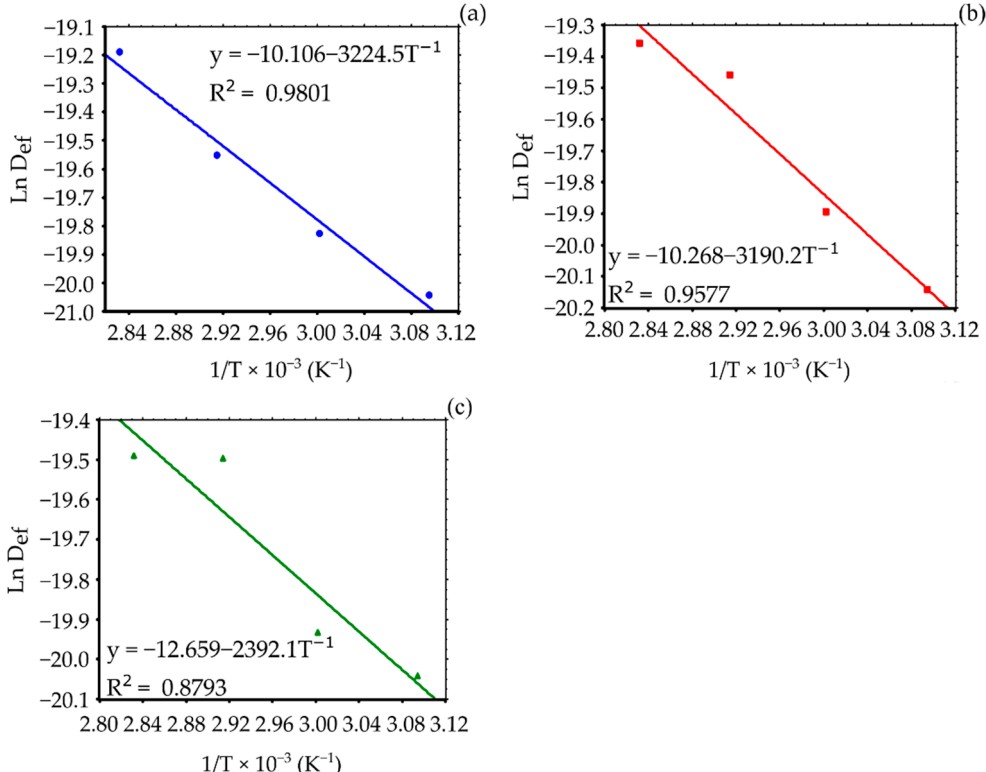

**Figure 3.** Arrhenius representation for the average effective diffusion coefficients obtained from drying germinated seeds of faba bean at temperatures of 50, 60, 70 and 80 °C for the varieties: (**a**) Orelha-de-Vó Preta; (**b**) Raio-de-Sol; and (**c**) Branca.

Table 5 shows the adjustment parameters of the Arrhenius equation for germinated seeds of faba beans. The activation energy to start the drying process of the germinated seeds of faba beans in the evaluated temperature range was similar for the OVP and RS varieties, and higher than the B variety, being all within the range described by Zogzas et al. [47], in which the activation energy for agricultural products can range from 12.7 to 110 kJ/mol. These values were higher than those reported by Ferreira et al. [43] in the drying of germinated pumpkin seeds, which ranged from 2.73 to 8.11 kJ/mol. These results indicate that the drying process of germinated faba bean seeds requires more energy for the diffusion of moisture to start.

**Table 5.** Fitting parameters of the Arrhenius equation for germinated faba bean seeds.

| Variety | $E_a$ (kJ/mol) | $D_0$ (m$^2$/s) | $R^2$ |
|---------|----------------|------------------|-------|
| OVP | 26.8085 | $4.0834 \times 10^{-5}$ | 0.9801 |
| RS | 26.5233 | $3.4727 \times 10^{-5}$ | 0.9577 |
| B | 19.8879 | $3.1788 \times 10^{-6}$ | 0.8783 |

OVP—Orelha-de-Vó Preta; RS—Raio-de-Sol; B—Branca; Ea—activation energy; $D_0$—pre-exponential factor; $R^2$—coefficient of determination.

### 3.3. Thermodynamic Properties

In Table 6, the average thermodynamic properties of dry germinated faba beans at different temperatures are presented. It is observed that the increase in the temperature of the drying air promotes a reduction in enthalpy ($\Delta H$), indicating, according to Morais et al. [48], that at higher temperatures there is less energy demand for the occurrence of dehydration of the samples. Furthermore, according to Shafaei et al. [49], positive enthalpy values indicate an endothermic process, that is, a process in which heat absorption occurs. According to Silva et al. [30], the reductions observed in entropy ($\Delta S$), with increasing temperature are related with the relative order of the system, where at lower temperatures there is less excitation of the water molecules, expressing, therefore, a greater degree of order. Negative entropy values can be attributed to the existence of structural changes in the adsorbent [50]. Gibbs free energy was directly proportional to the increase in temperature and showed positive values in the range evaluated. Positive values indicate an exogenous reaction, in which an external agent providing energy to the environment is needed for the reaction to occur, indicating a consistent result, since desorption is not a spontaneous reaction [49,51]. This thermodynamic property represents the maximum amount of energy released in a process under constant temperature and pressure that is available to be used, representing the balance between enthalpy and entropy [52]. Silva et al. [53] and Lisboa et al. [35] reported reductions in enthalpy and entropy and increases in Gibbs free energy as the drying temperature of soybeans was increased at 20, 30, 40 and 50 °C and of Mulatto beans (*Phaseolus vulgaris* L.) at temperatures of 40, 50, 60 and 70 °C.

### 3.4. Physicochemical Characterization

Table 7 shows the results of the physicochemical characterization of the germinated seeds of fresh faba beans and flours from the dry samples at temperatures of 50, 60, 70 and 80 °C. Drying reduced the water content of the samples to values between 1.55 and 5.18% at 80 and 50 °C, respectively. In all cases, these values are compatible with safe storage, with a reduction of about 96.34% when comparing the *in natura* and dried germinated seeds. Among the flours, the moisture content is below the upper limit recommended by the technical regulation for cassava flour [54] and wheat flour [55], which is 13 and 15%, respectively. Ferreira et al. [43] evaluating germinated pumpkin seeds (*Cucurbita moschata* D.), variety 'Jacarezinho' dried at 50, 60 and 70 °C, obtained water content values ranging from 1.10 to 5.93% (bs) being close to values obtained for the samples of the present work.

**Table 6.** Thermodynamic properties of germinated faba beans dried at temperatures of 50, 60, 70 and 80 °C.

| Variety | T (°C) | ΔH (kJ/mol) | ΔS (kJ/mol K) | ΔG (kJ/mol) |
|---------|--------|-------------|---------------|-------------|
| OVP | 50 | 24.1218 | −0.3296 | 130.6312 |
|  | 60 | 24.0387 | −0.3299 | 133.9284 |
|  | 70 | 23.9555 | −0.3301 | 137.2281 |
|  | 80 | 23.8724 | −0.3303 | 140.5303 |
| RS | 50 | 23.8367 | −0.3309 | 130.7812 |
|  | 60 | 23.7535 | −0.3312 | 134.0919 |
|  | 70 | 23.6704 | −0.3314 | 137.4051 |
|  | 80 | 23.5872 | −0.3317 | 140.7208 |
| B | 50 | 17.2013 | −0.3508 | 130.5696 |
|  | 60 | 17.1181 | −0.3511 | 134.0791 |
|  | 70 | 17.0350 | −0.3513 | 137.5911 |
|  | 80 | 16.9518 | −0.3516 | 141.1056 |

OVP—Orelha-de-Vó Preta; RS—Raio-de-Sol; B—Branca; ΔH—enthalpy; ΔS—entropy; ΔG—Gibbs free energy.

**Table 7.** Physicochemical parameters of fresh fava bean germinated seeds and flour from germinated samples dried at different temperatures.

| Drying Conditions | Water Content (g/100 g w.b.) | Water Activity (Decimal) | Ashes (g/100 g d.b.) | pH | Alcohol-Soluble Acidity (mL NaOH/100 g d.b.) | Total Sugars (g/100 g d.b.) | Reducing Sugars (g/100 g d.b.) | Crude Protein (g/100 g d.b.) | Starch (g/100 g d.b.) |
|---|---|---|---|---|---|---|---|---|---|
| OVP/FG | 52.73 ± 0.13 c | 0.996 ± 0.00 a | 4.11 ± 0.17 cde | 4.25 ± 0.02 h | 3.90 ± 0.00 c | 4.50 ± 0.34 i | 2.47 ± 0.04 h | 22.53 ± 0.85 ab | 45.41 ± 0.22 h |
| OVP/50 | 4.70 ± 0.07 e | 0.211 ± 0.00 d | 4.26 ± 0.10 cd | 4.44 ± 0.02 g | 0.58 ± 0.00 e | 10.54 ± 0.04 e | 2.26 ± 0.02 i | 24.62 ± 0.65 a | 54.35 ± 0.80 bcd |
| OVP/60 | 3.98 ± 0.05 f | 0.169 ± 0.01 f | 4.41 ± 0.16 abc | 4.48 ± 0.01 f | 0.38 ± 0.00 g | 10.49 ± 0.05 e | 2.29 ± 0.02 i | 24.07 ± 1.35 a | 52.36 ± 1.74 def |
| OVP/70 | 2.69 ± 0.08 hi | 0.130 ± 0.00 h | 4.69 ± 0.09 ab | 4.49 ± 0.01 f | 0.38 ± 0.00 ij | 10.22 ± 0.07 ef | 2.35 ± 0.03 hi | 22.52 ± 0.50 ab | 55.31 ± 0.67 b |
| OVP/80 | 2.53 ± 0.07 i | 0.119 ± 0.00 i | 4.80 ± 0.05 a | 4.50 ± 0.02 f | 0.38 ± 0.00 jl | 9.85 ± 0.11 f | 2.40 ± 0.01 hi | 23.74 ± 0.87 a | 49.75 ± 0.67 g |
| RS/FG | 56.27 ± 0.21 b | 0.998 ± 0.00 a | 4.12 ± 0.13 cde | 4.28 ± 0.02 h | 4.21 ± 0.00 b | 6.79 ± 0.03 g | 3.20 ± 0.10 fg | 23.20 ± 0.73 a | 55.58 ± 0.30 b |
| RS/50 | 4.22 ± 0.06 f | 0.244 ± 0.00 b | 3.85 ± 0.03 ef | 4.60 ± 0.01 de | 0.38 ± 0.00 g | 12.28 ± 0.06 c | 3.16 ± 0.01 g | 13.83 ± 0.64 d | 53.00 ± 0.75 cde |
| RS/60 | 3.23 ± 0.06 g | 0.199 ± 0.00 e | 3.56 ± 0.15 f | 4.70 ± 0.01 c | 0.38 ± 0.00 h | 12.10 ± 0.11 c | 3.37 ± 0.07 e | 10.71 ± 0.38 e | 52.66 ± 0.60 de |
| RS/70 | 2.32 ± 0.06 ij | 0.146 ± 0.00 g | 3.89 ± 0.11 def | 4.80 ± 0.01 b | 0.38 ± 0.00 jl | 11.72 ± 0.06 d | 3.89 ± 0.02 b | 9.89 ± 0.64 e | 55.06 ± 0.67 bc |
| RS/80 | 1.55 ± 0.04 l | 0.118 ± 0.00 i | 3.91 ± 0.13 def | 4.84 ± 0.01 a | 0.37 ± 0.00 m | 11.66 ± 0.12 h | 4.20 ± 0.05 a | 13.91 ± 0.23 d | 50.17 ± 0.96 fg |
| B/FG | 62.64 ± 0.21 a | 0.996 ± 0.00 a | 4.20 ± 0.12 cde | 4.27 ± 0.03 h | 4.91 ± 0.00 a | 6.26 ± 0.04 h | 3.65 ± 0.07 cd | 25.35 ± 0.63 a | 61.88 ± 0.32 a |
| B/50 | 5.18 ± 0.02 d | 0.222 ± 0.00 c | 4.37 ± 0.02 bc | 4.46 ± 0.02 fg | 0.58 ± 0.00 d | 14.27 ± 0.20 a | 3.75 ± 0.02 bc | 8.40 ± 0.45 e | 51.15 ± 0.69 efg |
| B/60 | 4.06 ± 0.02 f | 0.193 ± 0.00 e | 4.39 ± 0.15 bc | 4.56 ± 0.02 e | 0.58 ± 0.00 f | 13.90 ± 0.08 a | 3.58 ± 0.04 d | 10.04 ± 0.66 e | 53.43 ± 0.00 bcd |
| B/70 | 3.00 ± 0.04 gh | 0.135 ± 0.00 h | 4.08 ± 0.21 cde | 4.58 ± 0.01 de | 0.38 ± 0.00 hi | 13.25 ± 0.04 b | 3.32 ± 0.06 ef | 19.47 ± 0.48 c | 49.98 ± 0.68 g |
| B/80 | 2.14 ± 0.05 j | 0.111 ± 0.00 i | 4.07 ± 0.11 cd e | 4.62 ± 0.00 d | 0.38 ± 0.00 l | 10.41 ± 0.19 e | 3.42 ± 0.02 e | 19.84 ± 0.78 bc | 49.54 ± 0.67 g |

OVP/FG—orelha-de-vó preta fresh germinated; OVP/50—orelha-de-vó preta dry at 50 °C; OVP/60—orelha-de-vó preta dry at 60 °C; OVP/70—orelha-de-vó preta dry at 70 °C; OVP/80—orelha-de-vó preta dry at 80 °C; RS/FG—raio-de-sol fresh germinated; RS/50—raio-de-sol dry at 50 °C; RS/60—raio-de-sol dry at 60 °C; RS/70—raio-de-sol dry at 70 °C; RS/80—raio-de-sol dry at 80 °C; B/FG—branca fresh germinated; B/50—branca dry at 50 °C; B/60—branca dry at 60 °C; B/70—branca dry at 70 °C; B/80—branca dry at 80 °C. The values are means ± standard deviation of the determination in triplicate. Means with the same letter in the same column do not present statistical difference according to Tukey's test at 5% probability.

The values obtained for the water activity (Table 7) follow the behavior of the moisture content, decreasing with the increase in the drying temperature. Low values of $a_w$ contribute to the preservation of the product, as they reduce the availability of water for the proliferation of microorganisms and development of enzymatic reactions, favoring preservation and storage. Water activities lower than 0.8 reduce the development of bacteria

and below 0.6 reduce the development of fungi, yeast and mold [56]. Considering these values, it is concluded that the faba bean flours have a $a_w$ in the safety range against these agents at room temperature. Values like those for faba bean flour were found by Santos et al. [57] in red rice grain flours (*Oryza sativa* L.), dried at 40, 50, 60, 70 and 80 °C, obtained water activity values ranging from 0.101 to 0.229. Olagunju et al. [58] evaluated the water activity in ground, fermented and roasted bamboo flour (*Vigna underground* (L.) Green) during storage (lasting for 20 days) and obtained values ranging from 0.09 to 0.95, from 0.34 to 1.02 and from 0.42 to 0.89, respectively.

Ash contents ranged from 3.56 to 4.80%, with statistical equality between the fresh samples and the highest percentage in the 'orelha-de-vó' variety flour, followed by 'Branca'. Duenas et al. [59] evaluating black bean (4.3%) and pea (3.6%) seeds, both germinated for seven days at 20 °C, identified mean values of 4.3% and 3.6%, respectively, close to the ash content of the in natura bean samples. Ash values lower than those in the present study were quantified by Singh et al. [60], who reported 1.41, 1.27 and 1.16% for germinated soybean seed flours at temperatures of 25, 30 and 35 °C and dried at 45 °C in a convective dryer; and by Xu et al. [61], analyzing seed flours germinated for 72 and 96 h of chickpeas, lentils and yellow peas, which obtained mean ash values of 3.19 and 3.33%, 2.59 and 2.52% and 2.71 and 2.73%, respectively.

The samples presented an increase in pH as a function of the increase in drying temperature, remaining in the range of 4.44 to 4.84, similar to the in natura material. Variety RS had the highest values, followed by variety B and OVP. Higher values were identified by Silva et al. [62] on dried and freeze-dried alfalfa sprouts, pH 6.63 and 5.73, respectively; and by Santos et al. [57] who found values ranging from 6.72 to 6.79, for temperatures between 40 and 80 °C, in red rice flour. According to Kadam and Balasubramanian [63], pH below 4.5 lead to reduced growth of microorganisms, as is the case with the OVP sample flour, while samples with pH above 4.5 are prone to microbiological development and proliferation, in this case that flours of the RS and B varieties are included, with values slightly above this limit. An inverse behavior to the pH was observed for the alcohol-soluble acidity, which ranged from 0.37 to 0.58 mL NaOH/100 g (d.b.) in the flours, with lower results present in the RS variety flour (0.38–0.037 mL NaOH/100 g). The reduction in this parameter with drying may be related, according to Araújo et al. [64], to the oxidation of organic acids with increasing drying temperature. Reis et al. [65], studying the stability of the physicochemical properties of acerola flour, also observed a reduction in titratable acidity with increasing drying temperature. Contrary to Santos et al. [66] when evaluating the acidity of black rice grains, they observed an increase among the samples dried at temperatures between 40 and 80 °C.

The flours showed an increase in the content of total sugars in relation to the raw material and, among the flours, the increase in temperature promoted a gradual reduction in these. Lower results for total sugars were identified by Amadeu et al. [67] in kibbled seeds germinated for 48 h, of 5.67 g/100 g (d.b.), and for the flour of seeds germinated and dried at 70 °C, of 3.07 g/100 g (d.b.). Queiroz et al. [68] quantified for lychee seed flour the mean value of total sugars of 16.57 g/100 g (d.b.), therefore higher than those of fresh and dried faba bean samples. In the content of reducing sugars, the OVP variety maintains the relationship observed in total sugars, with values lower than the other varieties. Comparing the RS and B varieties, it is observed, as in the total sugars, an alternation between values, with statistically higher and lower results, indicating similarity between the samples. It is verified, with drying, by an increase in reducing sugars, with an increasing trend as the temperature of the drying air increases, except in the sample of variety B. The increase in the reducing sugar content with drying is commonly explained by the transformation of compounds into products of the Maillard reaction, which involves reactions of reducing sugars with amino groups [69]. Amadeu et al. [67] reported reducing sugar values of 5.97 and 1.59 g/100 g (d.b.) for germinated fresh pumpkin seeds and for seed flour, respectively. Moongngarm and Saetung [70] reported values for reducing sugars

of 10.9 g/100 g (d.b.) and totals of 14.6 g/100 g (d.b.) in germinated husk rice powder, exceeding those determined in faba bean flour.

Among the fresh samples, the Branca variety presented higher protein values than the OVP and RS, which have statistically similar values to each other. With drying, OVP maintained similar levels of protein between the fresh sample and the flours. Samples RS and B showed fluctuation in values, but with statistically significant reductions between the fresh sample and the flours. According to Driscoll [71], the reduction in protein content with increasing drying temperature can be explained by protein denaturation, with a decrease in its solubility as a result of higher temperatures. Duenas et al. [59] identified lower values for protein content, when evaluating the composition of germinated bean (*Phaseolus vulgaris* L.) and lentil (*Lens culinaris* L.) seed flours, being 15.7 and 18.9%, respectively. Xu et al. [61] obtained protein content of 26.06, 33.13 and 27.8 for chickpea (*Cicer aretinium* L.), lentil (*Lens culinaris* Merr.) and yellow pea (*Pisum sativum* L.) flours, respectively. According to the Brazilian Table of Food Composition (TACO) [72], the protein content of wheat and corn flour corresponds to 9.8 and 7.2%, respectively, indicating that the faba bean germinated, both in nature and in the form of flour, can adequately replace those traditionally consumed flours as a protein source.

The starch content among the samples germinated in natura was higher in the Branca variety, surpassing that of RS, which was higher than that of the OVP. Lower starch values than those found in natura faba beans were quantified by Xu et al. [61] evaluating, for six days, the starch content of chickpea, lentil, and yellow pea sprouts, all in nature, with values ranging from 38.51 to 43.81 g/100 g. It is observed that in the OVP sample the starch content of the flours was higher than in the fresh sample; while in the samples of the RS and B varieties, in general, the starch contents of the flours were lower than in the fresh samples. Starch values were reported by Cornejo et al. [73] in *Amaranthus quitensis* (black species) and *Amaranthus caudatus* (white species) flours, with mean values of 54.69 g/100 g and 27.07 g/100 g, respectively.

## 4. Conclusions

The Page and Midilli models presented the best adjustment parameters for the drying kinetics of the three faba bean varieties. The effective diffusivity coefficients were in the order of magnitude of values reported in the literature for this property ($10^{-11}$ to $10^{-9}$), increasing with increasing temperature and without significant differences between the varieties; the activation energies showed equivalent results in the Olho-de-Vó Preta and Raio-de-Sol varieties, surpassing the Branca variety by almost 30%, ranging from 19.89 to 26.81 kJ/mol. The entropy and enthalpy values were higher in the Olho-de-Vó Preta variety, followed by Raio-de-Sol, higher than in the Branca variety. The three varieties had approximate Gibbs free energy values; the increase in drying temperature resulted in a reduction in enthalpy and entropy and an increase in Gibbs free energy, corroborating that the drying process of germinated faba bean seeds is endothermic and requires external energy input. All samples showed acidic pH and the acidity was reduced with drying; the Raio-de-Sol and Branca varieties had higher sugar contents and the total sugars were increased with drying; the highest protein contents were determined in the Branca variety and in the in nature germinated samples; in the Branca variety the highest starch content was also verified.

**Author Contributions:** Conceptualization, L.T.S.A., A.J.d.M.Q. and R.M.F.d.F.; data curation, L.T.S.A., Y.F.P., H.V.M. and H.A.S.; formal analysis, W.P.d.S., J.P.G. and C.C.C.; investigation, J.P.d.L.F. and D.d.C.S.; methodology, L.T.S.A., A.J.d.M.Q. and J.P.d.L.F.; software, W.P.d.S. and J.P.G.; supervision, A.J.d.M.Q. and R.M.F.d.F.; validation, C.C.C., D.d.C.S. and A.R.C.d.L.; visualization, Y.F.P., H.V.M. and A.R.C.d.L.; writing—original draft, L.T.S.A.; writing—review and edit-ing, A.J.d.M.Q., R.M.F.d.F. and J.P.d.L.F.; funding acquisition, A.J.d.M.Q. All authors have read and agreed to the published version of the manuscript.

**Funding:** This research was funded by the Conselho Nacional de Desenvolvimento Científico e Tecno-lógico (CNPq): Process number 305972/2019-7 (Brazilian Research Agencie).

**Data Availability Statement:** Data can be digitized from the graphs or requested to the corresponding author.

**Acknowledgments:** The authors are grateful to the Federal University of Campina Grande (Brazil) for the research infrastructure.

**Conflicts of Interest:** The authors declare no conflict of interest.

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
