# Peer review of "Controlled Germination of Faba Beans: Drying, Thermodynamic Properties and Physical-Chemical Composition"

_processes, doi:10.3390/pr10081460_

Round 1
Reviewer 1 Report
I reviewed the manuscript and I have no suggestions regarding the applied methodology or gained results. This is a conventional approach for describing the drying kinetics and associated models fitting, and there is nothing wrong with it. Conclusions support the experimental results. However, I must say that there are many advanced modeling technics and methods in this scientific area that could be applied instead.
Author Response
Response to Reviewer #1 Comment
Dear reviewer #1
Firstly, thanks for your positive comments (and initial scores).
Reviewer(s)' Comments to Author:
I reviewed the manuscript and I have no suggestions regarding the applied methodology or gained results. This is a conventional approach for describing the drying kinetics and associated models fitting, and there is nothing wrong with it. Conclusions support the experimental results. However, I must say that there are many advanced modeling technics and methods in this scientific area that could be applied instead.
Answer: We would like to thank you for your comment.
Sincerely,
Corresponding author
Reviewer 2 Report
The introduction has described the background and the urgency of the research well.
The spacing style from line 99 to 108 need to adjust as per the MDPI journal format.
Line 177 to 180 were in MDPI journal format; the authors must remove these lines strictly.
Table 2 has been informative, but we need to add space between R2, MSD, and X2.
The results have been exceptionally well presented and well concluded.
We need to revise the similarity indicated by plagiarism strictly. The similarity index was 26% by Turnitin
Author Response
Response to Reviewer #2 Comments
Dear reviewer #2
We would like to thank you for your thoughtful comments, improving our manuscript.
Note: Besides the alterations explicitly mentioned, other modifications have been made (in red). The line numbers provided are referring to docx file with the new version of the article.
Reviewer: 2
Comments to the Author
The introduction has described the background and the urgency of the research well.
Answer: Firstly, thanks for your positive comments (and initial scores).
The spacing style from line 99 to 108 need to adjust as per the MDPI journal format.
Answer: Thanks for your observation. The text has been corrected (lines 94-103).
Line 177 to 180 were in MDPI journal format; the authors must remove these lines strictly.
Answer: We agree with reviewer #2 observation. The text has been corrected (lines 172-173).
Table 2 has been informative, but we need to add space between R2, MSD, and X2.
Answer: Thanks for your observation. The Table (located between lines 186-190) has been modified and we believe they are better now.
The results have been exceptionally well presented and well concluded.
Answer: Thanks for your positive comment.
We need to revise the similarity indicated by plagiarism strictly. The similarity index was 26% by Turnitin.
Answer: We believe the article is better now than in its old version, but if the reviewer #2 think that new modifications are still necessary, we are ready to make them.
Sincerely,
Corresponding author
Reviewer 3 Report
The manuscript presents a practical view about an agricultural crop that is widely used among consumers. Authors have followed the initial design of their study in the interpretation of their results. The manuscript has several minor language mistakes that need the attention of the authors.
The Abstract could be a bit shorter, with no background about the studied beans.
Author Response
Response to Reviewer #3 Comments
Dear reviewer #3
We would like to thank you for your thoughtful comments, improving our manuscript.
Note: Besides the alterations explicitly mentioned, other modifications have been made (in red). The line numbers provided are referring to docx file with the new version of the article.
Reviewer: 3
Comments to the Author
The manuscript presents a practical view about an agricultural crop that is widely used among consumers. Authors have followed the initial design of their study in the interpretation of their results. The manuscript has several minor language mistakes that need the attention of the authors.
Answer: Firstly, thanks for your positive comments (and initial scores). The manuscript had the language errors revised and hope the manuscript is better now.
The Abstract could be a bit shorter, with no background about the studied beans.
Answer: Thank you for your comments. The text has been corrected (lines 21-39).
Sincerely,
Corresponding author